# Understanding the Limitations of Variational Mutual Information Estimators

**Jiaming Song & Stefano Ermon**
Stanford University
{tsong, ermon}@cs.stanford.edu

## Abstract

Variational approaches based on neural networks are showing promise for estimating mutual information (MI) between high dimensional variables. However, they can be difficult to use in practice due to poorly understood bias/variance tradeoffs. We theoretically show that, under some conditions, estimators such as MINE exhibit variance that could grow exponentially with the true amount of underlying MI. We also empirically demonstrate that existing estimators fail to satisfy basic self-consistency properties of MI, such as data processing and additivity under independence. Based on a unified perspective of variational approaches, we develop a new estimator that focuses on variance reduction. Empirical results on standard benchmark tasks demonstrate that our proposed estimator exhibits improved bias-variance trade-offs on standard benchmark tasks.

## 1 Introduction

Mutual information (MI) estimation and optimization are crucial to many important problems in machine learning, such as representation learning (Chen et al., 2016; Zhao et al., 2018b; Tishby & Zaslavsky, 2015; Higgins et al., 2018) and reinforcement learning (Pathak et al., 2017; van den Oord et al., 2018). However, estimating mutual information from samples is challenging (McAllester & Statos, 2018) and traditional parametric and non-parametric approaches (Nemenman et al., 2004; Gao et al., 2015; 2017) struggle to scale up to modern machine learning problems, such as estimating the MI between images and learned representations.

Recently, there has been a surge of interest in MI estimation with variational approaches (Barber & Agakov, 2003; Nguyen et al., 2010; Donsker & Varadhan, 1975), which can be naturally combined with deep learning methods (Alemi et al., 2016; van den Oord et al., 2018; Poole et al., 2019). Despite their empirical effectiveness in downstream tasks such as representation learning (Hjelm et al., 2018; Veličković et al., 2018), their effectiveness *for MI estimation* remains unclear. In particular, higher estimated MI between observations and learned representations do not seem to indicate improved predictive performance when the representations are used for downstream supervised learning tasks (Tschannen et al., 2019).

In this paper, we discuss two limitations of variational approaches to MI estimation. First, we theoretically demonstrate that the variance of certain estimators, such as MINE (Belghazi et al., 2018), could grow *exponentially* with the ground truth MI, leading to poor bias-variance trade-offs. Second, we propose a set of *self-consistency tests* over basic properties of MI, and empirically demonstrate that all considered variational estimators fail to satisfy critical properties of MI, such as data processing and additivity under independence. These limitations challenge the effectiveness of these methods for estimating or optimizing MI.

To mitigate these issues, we propose a unified perspective over variational estimators treating variational MI estimation as an optimization problem over (valid) density ratios. This view highlights the role of partition functions estimation, which is the culprit of high variance issues in MINE. To address this issue, we propose to improve MI estimation via variance reduction techniques for partition function estimation. Empirical results demonstrate that our estimators have much better bias-variance trade-off compared to existing methods on standard benchmark tasks.

## 2    BACKGROUND AND RELATED WORK

### 2.1    NOTATIONS

We use uppercase letters to denote a probability measure (e.g., $P$, $Q$) and corresponding lowercase letters to denote its density[1] functions (e.g., $p$, $q$) unless specified otherwise. We use $X, Y$ to denote random variables with *separable* sample spaces denoted as $\mathcal{X}$ and $\mathcal{Y}$ respectively, and $\mathcal{P}(\mathcal{X})$ (or $\mathcal{P}(\mathcal{Y})$) to denote the set of all probability measures over the Borel $\sigma$-algebra on $\mathcal{X}$ (or $\mathcal{Y}$).

Under $Q \in \mathcal{P}(\mathcal{X})$, the $p$-norm of a function $r : \mathcal{X} \to \mathbb{R}$ is defined as $\|r\|_p := (\int |r|^p \mathrm{d}Q)^{1/p}$ with $\|r\|_\infty = \lim_{p \to \infty} \|r\|_p$. The set of locally $p$-integrable functions is defined as $L^p(Q) := \{r : \mathcal{X} \to \mathbb{R} \mid \|r\|_p < \infty\}$. The space of probability measures wrt. $Q$ is defined as $\Delta(Q) := \{r \in L^1(Q) \mid \|r\|_1 = 1, r \geq 0\}$; we also call this the space of "valid density ratios" wrt. $Q$. We use $P \ll Q$ to denote that $P$ is absolutely continuous with respect to $Q$. We use $\hat{I}_E$ to denote an estimator for $I_E$ where we replace expectations with sample averages.

### 2.2    VARIATIONAL MUTUAL INFORAMTION ESTIMATION

The mutual information between two random variables $X$ and $Y$ is the KL divergence between the joint and the product of marginals:

$$I(X;Y) = D_{\mathrm{KL}}(P(X,Y)\|P(X)P(Y)) \tag{1}$$

which we wish to estimate using samples from $P(X,Y)$; in certain cases we may know the density of marginals (e.g. $P(X)$). There are a wide range of variational approaches to variational MI estimation. Variational information maximization uses the following result (Barber & Agakov, 2003):

**Lemma 1** (Barber-Agakov (BA)). *For two random variables $X$ and $Y$:*

$$I(X;Y) = \sup_{q_\phi} \left\{ \mathbb{E}_{P(X,Y)} \left[ \log q_\phi(\boldsymbol{x}|\boldsymbol{y}) - \log p(\boldsymbol{x}) \right] =: I_{\mathrm{BA}}(q_\phi) \right\} \tag{2}$$

*where $q_\phi : \mathcal{Y} \to \mathcal{P}(\mathcal{X})$ is a valid conditional distribution over $\mathcal{X}$ given $\boldsymbol{y} \in \mathcal{Y}$ and $p(\boldsymbol{x})$ is the probability density function of the marginal distribution $P(X)$.*

Another family of approaches perform MI estimation through variational lower bounds to KL divergences. For example, the Mutual Information Neural Estimator (MINE, Belghazi et al. (2018)) applies the following lower bound to KL divergences (Donsker & Varadhan, 1975).

**Lemma 2** (Donsker-Varadahn (DV)). *$\forall P, Q \in \mathcal{P}(\mathcal{X})$ such that $P \ll Q$,*

$$D_{\mathrm{KL}}(P\|Q) = \sup_{T \in L^\infty(Q)} \left\{ \mathbb{E}_P[T] - \log \mathbb{E}_Q[e^T] =: I_{\mathrm{MINE}}(T) \right\}. \tag{3}$$

One could set $P = P(X,Y)$ and $Q = P(X)P(Y)$, $T$ as a parametrized neural network (e.g. $T_\theta(\boldsymbol{x}, \boldsymbol{y})$ parametrized by $\theta$), and obtain the estimate by optimizing the above objective via stochastic gradient descent over mini-batches. However, the corresponding estimator $\hat{I}_{\mathrm{MINE}}$ (where we replace the expectations in Eq. (3) with sample averages) is biased, leading to biased gradient estimates; Belghazi et al. (2018) propose to reduce bias via estimating the partition function $\mathbb{E}_Q[e^T]$ with exponential moving averages of mini-batches.

The variational $f$-divergence estimation approach (Nguyen et al., 2010; Nowozin et al., 2016) considers lower bounds on $f$-divergences which can be specialize to KL divergence, and subsequently to mutual information estimation:

**Lemma 3** (Nyugen et al. (NWJ)). *$\forall P, Q \in \mathcal{P}(\mathcal{X})$ such that $P \ll Q$,*

$$D_{\mathrm{KL}}(P\|Q) = \sup_{T \in L^\infty(Q)} \left\{ \mathbb{E}_P[T] - \mathbb{E}_Q[e^{T-1}] =: I_{\mathrm{NWJ}}(T) \right\} \tag{4}$$

*and $D_{\mathrm{KL}}(P\|Q) = I_{\mathrm{NWJ}}(T)$ when $T = \log(\mathrm{d}P/\mathrm{d}Q) + 1$.*

---

[1]In the remainder of the paper, we slightly overload "density" for discrete random variables.

The supremum over $T$ is a invertible function of the density ratio $\mathrm{d}P/\mathrm{d}Q$, so one could use this approach to estimate density ratios by inverting the function (Nguyen et al., 2010; Nowozin et al., 2016; Grover & Ermon, 2017). The corresponding mini-batch estimator (denoted as $\hat{I}_{\mathrm{NWJ}}$) is unbiased, so unlike MINE, this approach does not require special care to reduce bias in gradients.

Contrastive Predictive Coding (CPC, van den Oord et al. (2018)) considers the following objective:

$$I_{\mathrm{CPC}}(f_\theta) := \mathbb{E}_{P^n(X,Y)}\left[\frac{1}{n}\sum_{i=1}^n \log \frac{f_\theta(\boldsymbol{x}_i, \boldsymbol{y}_i)}{\frac{1}{n}\sum_{j=1}^n f_\theta(\boldsymbol{x}_i, \boldsymbol{y}_j)}\right] \tag{5}$$

where $f_\theta : \mathcal{X} \times \mathcal{Y} \to \mathbb{R}_{\geq 0}$ is a neural network parametrized by $\theta$ and $P^n(X,Y)$ denotes the joint pdf for $n$ i.i.d. random variables sampled from $P(X,Y)$. CPC generally has less variance but is more biased because its estimate does not exceed $\log n$, where $n$ is the batch size (van den Oord et al., 2018; Poole et al., 2019). While one can further reduce the bias with larger $n$, the number of evaluations needed for estimating each batch with $f_\theta$ is $n^2$, which scales poorly. To address the high-bias issue of CPC, Poole et al. proposed an interpolation between $I_{\mathrm{CPC}}$ and $I_{\mathrm{NWJ}}$ to obtain more fine-grained bias-variance trade-offs.

# 3 VARIATIONAL MUTUAL INFORMATION ESTIMATION AS OPTIMIZATION OVER DENSITY RATIOS

In this section, we unify several existing methods for variational mutual information estimation. We first show that variational mutual information estimation can be formulated as a constrained optimization problem, where the feasible set is $\Delta(Q)$, i.e. the valid density ratios with respect to $Q$.

**Theorem 1.** $\forall P, Q \in \mathcal{P}(\mathcal{X})$ such that $P \ll Q$ we have

$$D_{\mathrm{KL}}(P\|Q) = \sup_{r \in \Delta(Q)} \mathbb{E}_P[\log r] \tag{6}$$

where the supremum is achived when $r = \mathrm{d}P/\mathrm{d}Q$.

We defer the proof in Appendix A. The above argument works for KL divergence between general distributions, but in this paper we focus on the special case of mutual information estimation. For the remainder of the paper, we use $P$ to represent the short-hand notation for the joint distribution $P(X,Y)$ and use $Q$ to represent the short-hand notation for the product of marginals $P(X)P(Y)$.

## 3.1 A SUMMARY OF EXISTING VARIATIONAL METHODS

From Theorem 1, we can describe a general approach to variational MI estimation:

1. Obtain a density ratio estimate – denote the solution as $r$;
2. Project $r$ to be close to $\Delta(Q)$ – in practice we only have samples from $Q$, so we denote the solution as $\Gamma(r; Q_n)$, where $Q_n$ is the empirical distribution of $n$ i.i.d. samples from $Q$;
3. Estimate mutual information with $\mathbb{E}_P[\log \Gamma(r; Q_n)]$.

We illustrate two examples of variational mutual information estimation that can be summarized with this approach. In the case of Barber-Agakov, the proposed density ratio estimate is $r_{\mathrm{BA}} = q_\phi(\boldsymbol{x}|\boldsymbol{y})/p(\boldsymbol{x})$ (assuming that $p(\boldsymbol{x})$ is known), which is guaranteed to be in $\Delta(Q)$ because

$$\mathbb{E}_Q\left[q_\phi(\boldsymbol{x}|\boldsymbol{y})/p(\boldsymbol{x})\right] = \int q_\phi(\boldsymbol{x}|\boldsymbol{y})/p(\boldsymbol{x})\mathrm{d}P(x)\mathrm{d}P(y) = 1, \qquad \Gamma_{\mathrm{BA}}(r_{\mathrm{BA}}, Q_n) = r_{\mathrm{BA}} \tag{7}$$

for all conditional distributions $q_\phi$. In the case of MINE / Donsker-Varadahn, the *logarithm* of the density ratio is estimated with $T_\theta(\boldsymbol{x}, \boldsymbol{y})$; the corresponding density ratio might not be normalized, so one could apply the following normalization for $n$ samples:

$$\mathbb{E}_{Q_n}\left[e^{T_\theta}/\mathbb{E}_{Q_n}[e^{T_\theta}]\right] = 1, \qquad \Gamma_{\mathrm{MINE}}(e^{T_\theta}, Q_n) = e_\theta^T/\mathbb{E}_{Q_n}[e^{T_\theta}] \tag{8}$$

where $\mathbb{E}_{Q_n}[e^{T_\theta}]$ (the sample average) is an unbiased estimate of the partition function $\mathbb{E}_Q[e^{T_\theta}]$; $\Gamma_{\mathrm{DV}}(e^{T_\theta}, Q_n) \in \Delta(Q)$ is only true when $n \to \infty$. Similarly, we show $I_{\mathrm{CPC}}$ is a lower bound to MI in Corollary 2, Appendix A, providing an alternative proof to the one in Poole et al. (2019).

These examples demonstrate that different mutual information estimators can be obtained in a procedural manner by implementing the above steps, and one could involve different objectives at each step. For example, one could estimate density ratio via logistic regression (Hjelm et al., 2018; Poole et al., 2019; Mukherjee et al., 2019) while using $I_{\text{NWJ}}$ or $I_{\text{MINE}}$ to estimate MI. While logistic regression does not optimize for a lower bound for KL divergence, it provides density ratio estimates between $P$ and $Q$ which could be used for subsequent steps.

Table 1: Summarization of variational estimators of mutual information. The $\in \Delta(Q)$ column denotes whether the estimator is a valid density ratio wrt. $Q$. ($\checkmark$) means any parameterization is valid; ($n \to \infty$) means any parameterization is valid as the batch size grows to infinity; ($\text{tr} \to \infty$) means only the optimal parametrization is valid (infinite training cost).

| Category | Estimator | Params | $\Gamma(r; Q_n)$ | $\in \Delta(Q)$ |
|---|---|---|---|---|
| Gen. | $\hat{I}_{\text{BA}}$ | $q_\phi$ | $q_\phi(\boldsymbol{x}\|\boldsymbol{y})/p(\boldsymbol{x})$ | $\checkmark$ |
| | $\hat{I}_{\text{GM}}$ (Eq. (9)) | $p_\theta, p_\phi, p_\psi$ | $p_\theta(\boldsymbol{x}, \boldsymbol{y})/p_\phi(\boldsymbol{x})p_\psi(\boldsymbol{y})$ | $\text{tr} \to \infty$ |
| Disc. | $\hat{I}_{\text{MINE}}$ | $T_\theta$ | $e^{T_\theta(\boldsymbol{x},\boldsymbol{y})}/\mathbb{E}_{Q_n}[e^{T_\theta(\boldsymbol{x},\boldsymbol{y})}]$ | $n \to \infty$ |
| | $\hat{I}_{\text{CPC}}$ | $f_\theta$ | $f_\theta(\boldsymbol{x}, \boldsymbol{y})/\mathbb{E}_{P_n(Y)}[f_\theta(\boldsymbol{x}, \boldsymbol{y})]$ | $\checkmark$ |
| | $\hat{I}_{\text{SMILE}}$ (Eq. (17)) | $T_\theta, \tau$ | $e^{T_\theta(\boldsymbol{x},\boldsymbol{y})}/\mathbb{E}_{Q_n}[e^{\text{clip}(T_\theta(\boldsymbol{x},\boldsymbol{y}),-\tau,\tau)}]$ | $n, \tau \to \infty$ |

## 3.2 GENERATIVE AND DISCRIMINATIVE APPROACHES TO MI ESTIMATION

The above discussed variational mutual information methods can be summarized into two broad categories based on how the density ratio is obtained.

- The *discriminative approach* estimates the density ratio $\mathrm{d}P/\mathrm{d}Q$ directly; examples include the MINE, NWJ and CPC estimators.

- The *generative approach* estimates the densities of $P$ and $Q$ separately; examples include the BA estimator where a conditional generative model is learned. In addition, we describe a generative approach that explicitly learns generative models (GM) for $P(X, Y)$, $P(X)$ and $P(Y)$:

$$I_{\text{GM}}(p_\theta, p_\phi, p_\psi) := \mathbb{E}_P[\log p_\theta(\boldsymbol{x}, \boldsymbol{y}) - \log p_\phi(\boldsymbol{x}) - \log p_\psi(\boldsymbol{y})], \quad (9)$$

where $p_\theta, p_\phi, p_\psi$ are maximum likelihood estimates of $P(X, Y)$, $P(X)$ and $P(Y)$ respectively. We can learn the three distributions with generative models, such as VAE (Kingma & Welling, 2013) or Normalizing flows (Dinh et al., 2016), from samples.

We summarize various generative and discriminative variational estimators in Table 1.

**Differences between two approaches** While both *generative* and *discriminative* approaches can be summarized with the procedure in Section 3.1, they imply different choices in modeling, estimation and optimization.

- On the modeling side, the *generative approaches* might require more stringent assumptions on the architectures (e.g. likelihood or evidence lower bound is tractable), whereas the *discriminative approaches* do not have such restrictions.

- On the estimation side, *generative approaches* do not need to consider samples from the product of marginals $P(X)P(Y)$ (since it can model $P(X, Y)$, $P(X)$, $P(Y)$ separately), yet the *discriminative approaches* require samples from $P(X)P(Y)$; if we consider a mini-batch of size $n$, the number of evaluations for generative approaches is $\Omega(n)$ whereas that for discriminative approaches it could be $\Omega(n^2)$.

- On the optimization side, discriminative approaches may need additional projection steps to be close to $\Delta(Q)$ (such as $I_{\text{MINE}}$), while generative approaches might not need to perform this step (such as $I_{\text{BA}}$).

## 4 LIMITATIONS OF EXISTING VARIATIONAL ESTIMATORS

### 4.1 GOOD DISCRIMINATIVE ESTIMATORS REQUIRE EXPONENTIALLY LARGE BATCHES

In the $\hat{I}_{\text{NWJ}}$ and $\hat{I}_{\text{MINE}}$ estimators, one needs to estimate the "partition function" $\mathbb{E}_Q[r]$ for some density ratio estimator $r$; for example, $\hat{I}_{\text{MINE}}$ needs this in order to perform the projection step $\Gamma_{\text{MINE}}(r, Q_n)$ in Eq (8). Note that the $I_{\text{NWJ}}$ and $I_{\text{MINE}}$ lower bounds are maximized when $r$ takes the optimal value $r^\star = \mathrm{d}P/\mathrm{d}Q$. However, the sample averages $\hat{I}_{\text{MINE}}$ and $\hat{I}_{\text{NWJ}}$ of $\mathbb{E}_Q[r^\star]$ could have a variance that scales *exponentially* with the ground-truth MI; we show this in Theorem 2.

**Theorem 2.** *Assume that the ground truth density ratio $r^\star = \mathrm{d}P/\mathrm{d}Q$ and $\text{Var}_Q[r^\star]$ exist. Let $Q_n$ denote the empirical distribution of $n$ i.i.d. samples from $Q$ and let $\mathbb{E}_{Q_n}$ denote the sample average over $Q_n$. Then under the randomness of the sampling procedure, we have:*

$$\text{Var}_Q[\mathbb{E}_{Q_n}[r^\star]] \geq \frac{e^{D_{\text{KL}}(P\|Q)} - 1}{n} \tag{10}$$

$$\lim_{n\to\infty} n\text{Var}_Q[\log \mathbb{E}_{Q_n}[r^\star]] \geq e^{D_{\text{KL}}(P\|Q)} - 1. \tag{11}$$

We defer the proof to Appendix A. Note that in the theorem above, we assume the ground truth density ratio $r^\star$ is already obtained, which is the optimal ratio for NWJ and MINE estimators. As a natural consequence, the NWJ and MINE estimators under the optimal solution could exhibit variances that grow exponentially with the ground truth MI (recall that in our context MI is a KL divergence). One could achieve smaller variances with some $r \neq r^\star$, but this guarantees looser bounds and higher bias.

**Corollary 1.** *Assume that the assumptions in Theorem 2 hold. Let $P_m$ and $Q_n$ be the empirical distributions of $m$ i.i.d. samples from $P$ and $n$ i.i.d. samples from $Q$, respectively. Define*

$$I_{\text{NWJ}}^{m,n} := \mathbb{E}_{P_m}[\log r^\star + 1] - \mathbb{E}_{Q_n}[r^\star] \tag{12}$$

$$I_{\text{MINE}}^{m,n} := \mathbb{E}_{P_m}[\log r^\star] - \log \mathbb{E}_{Q_n}[r^\star] \tag{13}$$

*where $r^\star = \mathrm{d}P/\mathrm{d}Q$. Then under the randomness of the sampling procedure, we have $\forall m \in \mathbb{N}$:*

$$\text{Var}_{P,Q}[I_{\text{NWJ}}^{m,n}] \geq (e^{D_{\text{KL}}(P\|Q)} - 1)/n \tag{14}$$

$$\lim_{n\to\infty} n\text{Var}_{P,Q}[I_{\text{MINE}}^{m,n}] \geq e^{D_{\text{KL}}(P\|Q)} - 1. \tag{15}$$

This high variance phenomenon has been empirically observed in Poole et al. (2019) (Figure 3) for $\hat{I}_{\text{NWJ}}$ under various batch sizes, where the log-variance scales linearly with MI. We also demonstrate this in Figure 2 (Section 6.1). In order to keep the variance of $\hat{I}_{\text{MINE}}$ and $\hat{I}_{\text{NWJ}}$ relatively constant with growing MI, one would need a batch size of $n = \Theta(e^{D_{\text{KL}}(P\|Q)})$. $\hat{I}_{\text{CPC}}$ has small variance, but it would need $n \geq e^{D_{\text{KL}}(P\|Q)}$ to have small bias, as its estimations are bounded by $\log n$.

### 4.2 SELF-CONSISTENCY ISSUES FOR MUTUAL INFORMATION ESTIMATORS

If we consider $\mathcal{X}, \mathcal{Y}$ to be high-dimensional, estimation of mutual information becomes more difficult. The density ratio between $P(X,Y)$ and $P(X)P(Y)$ could be very difficult to estimate from finite samples without proper parametric assumptions (McAllester & Statos, 2018; Zhao et al., 2018a). Additionally, the exact value of mutual information is dependent on the definition of the sample space; given finite samples, whether the underlying random variable is assumed to be discrete or continuous will lead to different measurements of mutual information (corresponding to entropy and differential entropy, respectively).

In machine learning applications, however, we are often more interested in maximizing or minimizing mutual information (estimates), rather than estimating its exact value. For example, if an estimator is off by a constant factor, it would still be useful for downstream applications, even though it can be highly biased. To this end, we propose a set of *self-consistency* tests for any MI estimator $\hat{I}$, based on properties of mutual information:

    1. (Independence) if $X$ and $Y$ are independent, then $\hat{I}(X;Y) = 0$;

2. (Data processing) for all functions $g, h$, $\hat{I}(X; Y) \geq \hat{I}(g(X); h(Y))$ and $\hat{I}(X; Y) \approx \hat{I}([X, g(X)]; [Y, h(Y)])$ where $[\cdot, \cdot]$ denotes concatenation.

3. (Additivity) denote $X_1, X_2$ as independent random variables that have the same distribution as $X$ (similarly define $Y_1, Y_2$), then $\hat{I}([X_1, X_2]; [Y_1, Y_2]) \approx 2 \cdot \hat{I}(X, Y)$.

These properties holds under both entropy and differential entropy, so they do not depend on the choice of the sample space. While these conditions are necessary but obviously not sufficient for accurate mutual information estimation, we argue that satisfying them is highly desirable for applications such as representation learning (Chen et al., 2016) and information bottleneck (Tishby & Zaslavsky, 2015). Unfortunately, none of the MI estimators we considered above pass all the self-consistency tests when $X, Y$ are images, as we demonstrate below in Section 6.2. In particular, the *generative* approaches perform poorly when MI is low (failing in independence and data processing), whereas *discriminative* approaches perform poorly when MI is high (failing in additivity).

## 5 IMPROVED MI ESTIMATION VIA CLIPPED DENSITY RATIOS

To address the high-variance issue in the $I_{\text{NWJ}}$ and $I_{\text{MINE}}$ estimators, we propose to clip the density ratios when estimating the partition function. We define the following clip function:

$$\text{clip}(v, l, u) = \max(\min(v, u), l) \tag{16}$$

For an empirical distribution of $n$ samples $Q_n$, instead of estimating the partition function via $\mathbb{E}_{Q_n}[r]$, we instead consider $\mathbb{E}_{Q_n}[\text{clip}(r, e^{-\tau}, e^{\tau})]$ where $\tau \geq 0$ is a hyperparameter; this is equivalent to clipping the log density ratio estimator between $-\tau$ and $\tau$.

We can then obtain a following estimator with smoothed partition function estimates:

$$I_{\text{SMILE}}(T_\theta, \tau) := \mathbb{E}_P[T_\theta(\boldsymbol{x}, \boldsymbol{y})] - \log \mathbb{E}_Q[\text{clip}(e^{T_\theta(\boldsymbol{x}, \boldsymbol{y})}, e^{-\tau}, e^{\tau})] \tag{17}$$

where $T_\theta$ is a neural network that estimates the log-density ratio (similar to the role of $T_\theta$ in $\hat{I}_{\text{MINE}}$). We term this the Smoothed Mutual Information "Lower-bound" Estimator (SMILE) with hyperparameter $\tau$; $I_{\text{SMILE}}$ converges to $I_{\text{MINE}}$ when $\tau \to \infty$. In our experiments, we consider learning the density ratio with logistic regression, similar to the procedure in Deep InfoMax (Hjelm et al., 2018).

The selection of $\tau$ affects the bias-variance trade-off when estimating the partition function; with a smaller $\tau$, variance is reduced at the cost of (potentially) increasing bias. In the following theorems, we analyze the bias and variance in the worst case for density ratio estimators whose actual partition function is $S$ for some $S \in (0, \infty)$.

**Theorem 3.** *Let $r(\boldsymbol{x}) : \mathcal{X} \to \mathbb{R}_{\geq 0}$ be any non-negative measurable function such that $\int r \, dQ = S$, $S \in (0, \infty)$ and $r(\boldsymbol{x}) \in [0, e^K]$. Define $r_\tau(\boldsymbol{x}) = \text{clip}(r(\boldsymbol{x}), e^\tau, e^{-\tau})$ for finite, non-negative $\tau$. If $\tau < K$, then the bias for using $r_\tau$ to estimate the partition function of $r$ satisfies:*

$$|\mathbb{E}_Q[r] - \mathbb{E}_Q[r_\tau]| \leq \max \left( e^{-\tau} |1 - Se^{-\tau}|, \left| \frac{1 - e^K e^{-\tau} + S(e^K - e^\tau)}{e^K - e^{-\tau}} \right| \right) ;$$

*if $\tau \geq K$, then*

$$|\mathbb{E}_Q[r] - \mathbb{E}_Q[r_\tau]| \leq e^{-\tau}(1 - Se^{-K}).$$

**Theorem 4.** *The variance of the estimator $\mathbb{E}_{Q_n}[r_\tau]$ (using $n$ samples from $Q$) satisfies:*

$$\text{Var}[\mathbb{E}_{Q_n}[r_\tau]] \leq \frac{e^\tau - e^{-\tau}}{4n} \tag{18}$$

We defer the proofs to Appendix A. Theorems 3 and 4 suggest that as we decrease $\tau$, variance is decreased at the cost of potentially increasing bias. However, if $S$ is close to 1, then we could use small $\tau$ values to obtain estimators where both variance and bias are small. We further discuss the bias-variance trade-off for a fixed $r$ over changes of $\tau$ in Theorem 3 and Corollary 3.

# 6 EXPERIMENTS

## 6.1 BENCHMARKING ON MULTIVARIATE GAUSSIANS

First, we evaluate the performance of MI bounds on two toy tasks detailed in (Poole et al., 2019; Belghazi et al., 2018), where the ground truth MI is tractable. The first task (**Gaussian**) is where $(\boldsymbol{x}, \boldsymbol{y})$ are drawn from a 20-d Gaussian distribution with correlation $\rho$, and the second task (**Cubic**) is the same as **Gaussian** but we apply the transformation $\boldsymbol{y} \mapsto \boldsymbol{y}^3$. We consider three discriminative approaches ($I_{\mathrm{CPC}}, I_{\mathrm{NWJ}}, I_{\mathrm{SMILE}}$) and one generative approach ($I_{\mathrm{GM}}$). For the discriminative approaches, we consider the joint critic in (Belghazi et al., 2018) and the separate critic in (van den Oord et al., 2018). For $I_{\mathrm{GM}}$ we consider invertible flow models (Dinh et al., 2016). We train all models for 20k iterations, with the ground truth mutual information increasing by 2 per 4k iterations. More training details are included in Appendix B[2].

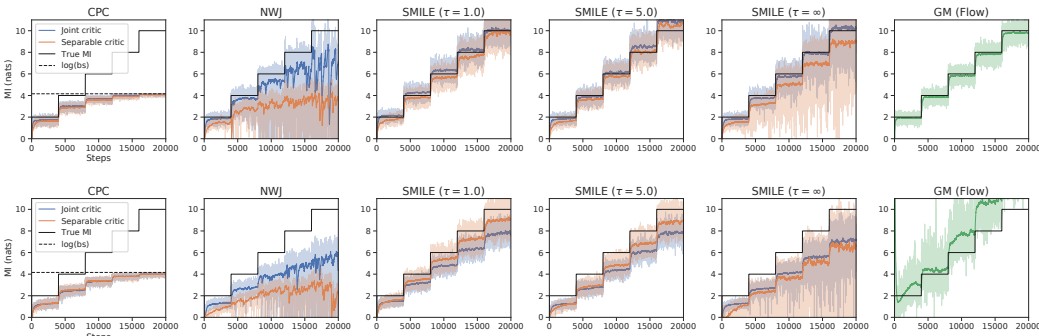

Figure 1: Performance of mutual information estimation approaches on **Gaussian** (top row) and **Cubic** (bottom row). Left two columns are $I_{\mathrm{CPC}}$ and $I_{\mathrm{NWJ}}$, next three columns are $I_{\mathrm{SMILE}}$ with $\tau = 1.0, 5.0, \infty$ and the right column is $I_{\mathrm{GM}}$ with flow models.

Figure 1 shows the estimated mutual information over the number of iterations. In both tasks, $I_{\mathrm{CPC}}$ has high bias and $I_{\mathrm{NWJ}}$ has high variance when the ground truth MI is high, whereas $I_{\mathrm{SMILE}}$ has relatively low bias and low variance across different architectures and tasks. Decreasing $\tau$ in the SMILE estimator decreases variances consistently but has different effects over bias; for example, under the joint critic bias is higher for $\tau = 5.0$ in **Gaussian** but lower in **Cubic**. $I_{\mathrm{GM}}$ with flow models has the best performance on **Gaussian**, yet performs poorly on **Cubic**, illustrating the importance of model parametrization in the *generative approaches*.

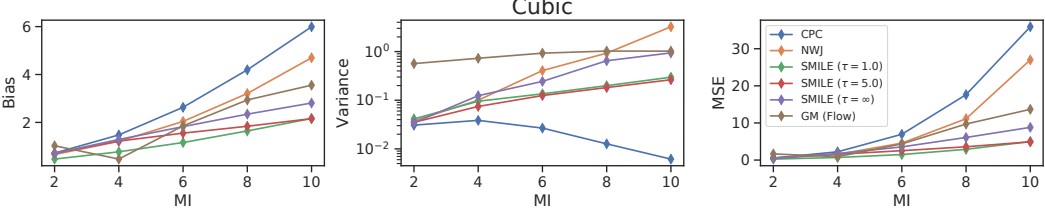

Figure 2: Bias / Variance / MSE of various estimators on **Cubic** (right). We display more results for **Gaussian** in Appendix B.

In Figure 2, we compare the bias, variance and mean squared error (MSE) of the *discriminative methods*. We observe that the variance of $I_{\mathrm{NWJ}}$ increases exponentially with mutual information, which is consistent with our theory in Corollary 1. On the other hand, the SMILE estimator is able to achieve much lower variances with small $\tau$ values; in comparison the variance of SMILE when $\tau = \infty$ is similar to that of $I_{\mathrm{NWJ}}$ in **Cubic**. In Table 2, we show that $I_{\mathrm{SMILE}}$ can have nearly two orders of magnitude smaller variance than $I_{\mathrm{NWJ}}$ while having similar bias. Therefore $I_{\mathrm{SMILE}}$ enjoys lower MSE in this benchmark MI estimation task compared to $I_{\mathrm{NWJ}}$ and $I_{\mathrm{CPC}}$.

---

[2]We release our code in

## 6.2 SELF-CONSISTENCY TESTS ON IMAGES

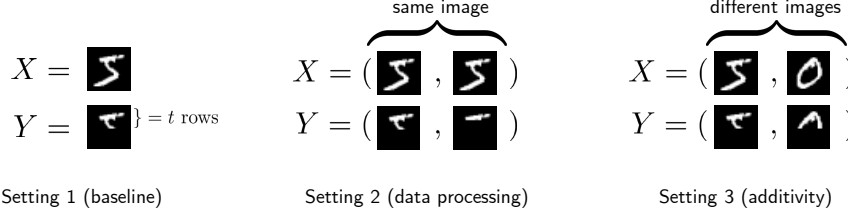

Figure 3: Three settings in the self-consistency experiments.

Next, we perform our proposed *self-consistency* tests on high-dimensional images (MNIST and CIFAR10) under three settings, where the ground truth MI is difficult to obtain (if not impossible). These settings are illustrated in Figure 3.

1. The first setting is where $X$ is an image and $Y$ is the same image where we mask the bottom rows, leaving the top $t$ rows from $X$ ($t$ is selected before evaluation). The rationale behind this choice of $Y$ is twofold: 1) $I(X;Y)$ should be non-decreasing with $t$; 2) it is easier (compared to low-d representations) to gain intuition about the amount of information remaining in $Y$.

2. In the second setting, $X$ corresponds to two identical images, and $Y$ to the top $t_1$, $t_2$ rows of the two images ($t_1 \geq t_2$); this considers the "data-processing" property.

3. In the third setting, $X$ corresponds to two independent images, and $Y$ to the top $t$ rows of both; this considers the "additivity" property.

We compare four approaches: $I_{\mathrm{CPC}}$, $I_{\mathrm{MINE}}$, $I_{\mathrm{SMILE}}$ and $I_{\mathrm{GM}}$. We use the same CNN architecture for $I_{\mathrm{CPC}}$, $I_{\mathrm{MINE}}$ and $I_{\mathrm{SMILE}}$, and use VAEs (Kingma & Welling, 2013) for $I_{\mathrm{GM}}$. We include more experimental details and alternative image processing approaches in Appendix B.

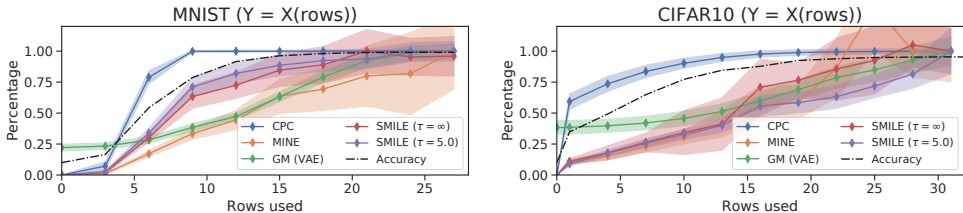

Figure 4: Evaluation of $\hat{I}(X;Y)/\hat{I}(X;,X)$. $X$ is an image and $Y$ contains the top $t$ rows of $X$.

**Baselines** We evaluate the first setting with $Y$ having varying number of rows $t$ in Figure 4, where the estimations are normalized by the estimated $\hat{I}(X;X)$. Most methods (except for $I_{\mathrm{GM}}$) predicts zero MI when $X$ and $Y$ are independent, passing the first self-consistency test. Moreover, the estimated MI is non-decreasing with increasing $t$, but with different slopes. As a reference, we show the validation accuracy of predicting the label where only the top $t$ rows are considered.

**Data-processing** In the second setting we set $t_2 = t_1 - 3$. Ideally, the estimator should satisfy $\hat{I}([X,X];[Y,h(Y)])/\hat{I}(X,Y) \approx 1$, as additional processing should not increase information. We show the above ratio in Figure 5 under varying $t_1$ values. All methods except for $I_{\mathrm{MINE}}$ and $I_{\mathrm{GM}}$ performs well in both datasets; $I_{\mathrm{GM}}$ performs poorly in CIFAR10 (possibly due to limited capacity of VAE), whereas $I_{\mathrm{MINE}}$ performs poorly in MNIST (possibly due to numerical stability issues).

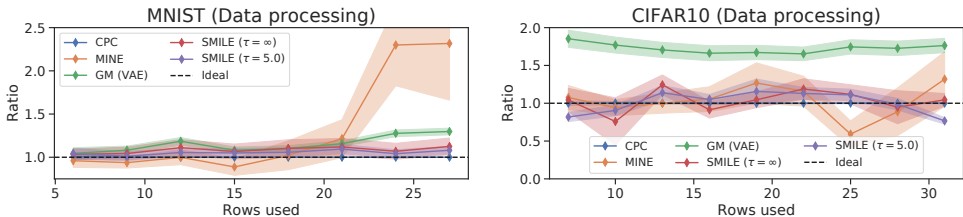

Figure 5: Evaluation of $\hat{I}([X, X]; [Y, h(Y)])/\hat{I}(X, Y)$, where the ideal value is 1.

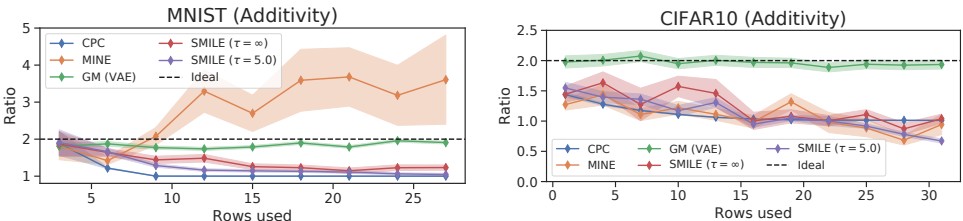

Figure 6: Evaluation of $\hat{I}([X_1, X_2]; [Y_1, Y_2])/\hat{I}(X, Y)$, where the ideal value is 2.

**Additivity** In the third setting, the estimator should double its value compared to the baseline with the same $t$, i.e. $\hat{I}([X_1, X_2]; [Y_1, Y_2])/\hat{I}(X, Y) \approx 2$. Figure 6 shows the above ratio under different values of $t$. None of the *discriminative* approaches worked well in this case except when $t$ is very small, when $t$ is large this ratio converges to 1 (possibly due to initialization and saturation of the training objective). $I_{\text{GM}}$ however, performs near perfectly on this test for all values of $t$.

## 7 DISCUSSION

In this work, we discuss *generative* and *discriminative* approaches to variational mutual information estimation and demonstrate their limitations. We show that estimators based on $I_{\text{NWJ}}$ and $I_{\text{MINE}}$ are prone to high variances when estimating with mini-batches, inspiring our $I_{\text{SMILE}}$ estimator that improves performances on benchmark tasks. However, none of the approaches are good enough to pass the *self-consistency* tests. The *generative* approaches perform poorly when MI is small (failing independence and data-processing tests) while the *discriminative* approaches perform poorly when MI is large (failing additivity tests).

These empirical evidences suggest that optimization over these variational estimators are not necessarily related to optimizing MI, so the empirical successes with these estimators might have little connections to optimizing mutual information. Therefore, it would be helpful to acknowledge these limitations and consider alternative measurements of information that are more suited for modern machine learning applications (Ozair et al., 2019; Tschannen et al., 2019).

### ACKNOWLEDGEMENTS

This research was supported by AFOSR (FA9550-19-1-0024), NSF (#1651565, #1522054, #1733686), ONR, and FLI. The authors would like to thank Shengjia Zhao, Yilun Xu and Lantao Yu for helpful discussions.

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

# A  PROOFS

## A.1  PROOFS IN SECTION 3

**Theorem 1.** $\forall P, Q \in \mathcal{P}(\mathcal{X})$ *such that* $P \ll Q$ *we have*

$$D_{\mathrm{KL}}(P\|Q) = \sup_{r \in \Delta(Q)} \mathbb{E}_P[\log r] \tag{6}$$

*where the supremum is achived when* $r = \mathrm{d}P/\mathrm{d}Q$.

*Proof.* For every $T \in L^\infty(Q)$, define $r_T = \frac{e^T}{\mathbb{E}_Q[e^T]}$, then $r_T \in \Delta(Q)$ and from the Donsker-Varadhan inequality (Donsker & Varadhan, 1975)

$$D_{\mathrm{KL}}(P\|Q) = \sup_{T \in L^\infty(Q)} \mathbb{E}_P[T] - \log \mathbb{E}_Q[e^T] \tag{19}$$

$$= \sup_{T \in L^\infty(Q)} \mathbb{E}_P\left[\log \frac{e^T}{\mathbb{E}_Q[e^T]}\right] = \sup_{r_T \in \Delta(Q)} \mathbb{E}_P[\log r_T] \tag{20}$$

Moreover, we have:

$$D_{\mathrm{KL}}(P\|Q) = \mathbb{E}_P[\log \mathrm{d}P - \log \mathrm{d}Q] = \mathbb{E}_P\left[\log \frac{\mathrm{d}P}{\mathrm{d}Q}\right] \tag{21}$$

which completes the proof. $\qquad\square$

**Corollary 2.** $\forall P, Q \in \mathcal{P}(\mathcal{X})$ *such that* $P \ll Q, \forall f_\theta : \mathcal{X} \to \mathbb{R}_{\geq 0}$ *we have*

$$I(X;Y) \geq I_{\mathrm{CPC}}(f_\theta) := \mathbb{E}_{P^n(X,Y)}\left[\frac{1}{n}\sum_{i=1}^n \log \frac{f_\theta(\boldsymbol{x}_i, \boldsymbol{y}_i)}{\frac{1}{n}\sum_{j=1}^n f_\theta(\boldsymbol{x}_i, \boldsymbol{y}_j)}\right] \tag{22}$$

*Proof.*

$$nI_{\mathrm{CPC}}(f_\theta) := \mathbb{E}_{P^n(X,Y)}\left[\sum_{i=1}^n \log \frac{f_\theta(\boldsymbol{x}_i, \boldsymbol{y}_i)}{\frac{1}{n}\sum_{j=1}^n f_\theta(\boldsymbol{x}_i, \boldsymbol{y}_j)}\right] \tag{23}$$

$$= \mathbb{E}_{P^n(X,Y)}\left[\sum_{i=1}^n \log \frac{n f_\theta(\boldsymbol{x}_i, \boldsymbol{y}_i)}{\sum_{j=1}^n f_\theta(\boldsymbol{x}_i, \boldsymbol{y}_j)}\right] \tag{24}$$

Since

$$\mathbb{E}_{P(X)P^n(Y)}\left[\frac{n f_\theta(\boldsymbol{x}, \boldsymbol{y})}{\sum_{j=1}^n f_\theta(\boldsymbol{x}, \boldsymbol{y}_j)}\right] = 1, \tag{25}$$

we can apply Theorem 1 to obtain:

$$nI_{\mathrm{CPC}}(f_\theta) = \mathbb{E}_{P^n(X,Y)}\left[\sum_{i=1}^n \log \frac{n f_\theta(\boldsymbol{x}_i, \boldsymbol{y}_i)}{\sum_{j=1}^n f_\theta(\boldsymbol{x}_i, \boldsymbol{y}_j)}\right] \tag{26}$$

$$= \sum_{i=1}^n \mathbb{E}_{P(X_i, Y_1^n)}\left[\log \frac{n f_\theta(\boldsymbol{x}_i, \boldsymbol{y}_i)}{\sum_{j=1}^n f_\theta(\boldsymbol{x}_i, \boldsymbol{y}_j)}\right] \tag{27}$$

$$\leq \sum_{i=1}^n I(X_i; Y_1^n) = nI(X;Y) \tag{28}$$

where $Y_1^n$ denotes the concatenation of $n$ independent random variables $(Y_1, \ldots, Y_n)$ and

$$P(X_i, Y_1^n) = P(X_i, Y_i)P(Y_1^{i-1})P(Y_{i+1}^n)$$

is the joint distribution of $P(X_i, Y_1^n)$. $\qquad\square$

A.2   PROOFS IN SECTION 4

**Theorem 2.** *Assume that the ground truth density ratio $r^\star = \mathrm{d}P/\mathrm{d}Q$ and $\mathrm{Var}_Q[r^\star]$ exist. Let $Q_n$ denote the empirical distribution of $n$ i.i.d. samples from $Q$ and let $\mathbb{E}_{Q_n}$ denote the sample average over $Q_n$. Then under the randomness of the sampling procedure, we have:*

$$\mathrm{Var}_Q[\mathbb{E}_{Q_n}[r^\star]] \geq \frac{e^{D_{\mathrm{KL}}(P\|Q)} - 1}{n} \tag{10}$$

$$\lim_{n\to\infty} n\mathrm{Var}_Q[\log \mathbb{E}_{Q_n}[r^\star]] \geq e^{D_{\mathrm{KL}}(P\|Q)} - 1. \tag{11}$$

*Proof.* Consider the variance of $r^\star(\boldsymbol{x})$ when $\boldsymbol{x} \sim Q$:

$$\mathrm{Var}_Q[r^\star] = \mathbb{E}_Q\left[\left(\frac{\mathrm{d}P}{\mathrm{d}Q}\right)^2\right] - \left(\mathbb{E}_Q\left[\frac{\mathrm{d}P}{\mathrm{d}Q}\right]\right)^2 \tag{29}$$

$$= \mathbb{E}_P\left[\frac{\mathrm{d}P}{\mathrm{d}Q}\right] - 1 \tag{30}$$

$$\geq e^{\mathbb{E}_P[\log \frac{\mathrm{d}P}{\mathrm{d}Q}]} - 1 \tag{31}$$

$$= e^{D_{\mathrm{KL}}(P\|Q)} - 1 \tag{32}$$

where (29) uses the definition of variance, (30) uses the definition of Radon-Nikodym derivative to change measures, (31) uses Jensen's inequality over log, and (32) uses the definition of KL divergences.

The variance of the mean of $n$ i.i.d. random variables then gives us:

$$\mathrm{Var}_Q[\mathbb{E}_{Q_n}[r]] = \frac{\mathrm{Var}[r]}{n} \geq \frac{e^{D_{\mathrm{KL}}(P\|Q)} - 1}{n} \tag{33}$$

which is the first part of the theorem.

As $n \to \infty$, $\mathrm{Var}_Q[\mathbb{E}_{Q_n}[r]] \to 0$, so we can apply the delta method:

$$\mathrm{Var}_Q[f(X)] \approx (f'(\mathbb{E}(X)))^2 \mathrm{Var}_Q[X] \tag{34}$$

Applying $f = \log$ and $\mathbb{E}[X] = 1$ gives us the second part of the theorem:

$$\lim_{n\to\infty} n\mathrm{Var}_Q[\log \mathbb{E}_{Q_n}[r]] = \lim_{n\to\infty} n\mathrm{Var}[\mathbb{E}_{Q_n}[r]] \geq e^{D_{\mathrm{KL}}(P\|Q)} - 1 \tag{35}$$

which describes the variance in the asymptotic sense. □

**Corollary 1.** *Assume that the assumptions in Theorem 2 hold. Let $P_m$ and $Q_n$ be the empirical distributions of $m$ i.i.d. samples from $P$ and $n$ i.i.d. samples from $Q$, respectively. Define*

$$I_{\mathrm{NWJ}}^{m,n} := \mathbb{E}_{P_m}[\log r^\star + 1] - \mathbb{E}_{Q_n}[r^\star] \tag{12}$$

$$I_{\mathrm{MINE}}^{m,n} := \mathbb{E}_{P_m}[\log r^\star] - \log \mathbb{E}_{Q_n}[r^\star] \tag{13}$$

*where $r^\star = \mathrm{d}P/\mathrm{d}Q$. Then under the randomness of the sampling procedure, we have $\forall m \in \mathbb{N}$:*

$$\mathrm{Var}_{P,Q}[I_{\mathrm{NWJ}}^{m,n}] \geq (e^{D_{\mathrm{KL}}(P\|Q)} - 1)/n \tag{14}$$

$$\lim_{n\to\infty} n\mathrm{Var}_{P,Q}[I_{\mathrm{MINE}}^{m,n}] \geq e^{D_{\mathrm{KL}}(P\|Q)} - 1. \tag{15}$$

*Proof.* Since $P_m$ and $Q_n$ are independent, we have

$$\mathrm{Var}[I_{\mathrm{NWJ}}^{m,n}] \geq \mathrm{Var}[\mathbb{E}_{Q_n}[r^\star]] \tag{36}$$

$$= \mathrm{Var}[\mathbb{E}_{Q_n}[r^\star]] \geq \frac{e^{D_{\mathrm{KL}}(P\|Q)} - 1}{n} \tag{37}$$

and

$$\lim_{n\to\infty} n\mathrm{Var}[I_{\mathrm{MINE}}^{m,n}] \geq \lim_{n\to\infty} n\mathrm{Var}[\log \mathbb{E}_{Q_n}[r^\star]] \geq e^{D_{\mathrm{KL}}(P\|Q)} - 1 \tag{38}$$

which completes the proof. □

### A.3 PROOFS IN SECTION 5

**Theorem 3.** *Let $r(\boldsymbol{x}) : \mathcal{X} \to \mathbb{R}_{\geq 0}$ be any non-negative measurable function such that $\int r \mathrm{d}Q = S$, $S \in (0, \infty)$ and $r(\boldsymbol{x}) \in [0, e^K]$. Define $r_\tau(\boldsymbol{x}) = \mathrm{clip}(r(\boldsymbol{x}), e^\tau, e^{-\tau})$ for finite, non-negative $\tau$. If $\tau < K$, then the bias for using $r_\tau$ to estimate the partition function of $r$ satisfies:*

$$|\mathbb{E}_Q[r] - \mathbb{E}_Q[r_\tau]| \leq \max\left(e^{-\tau}|1 - Se^{-\tau}|, \left|\frac{1 - e^K e^{-\tau} + S(e^K - e^\tau)}{e^K - e^{-\tau}}\right|\right);$$

*if $\tau \geq K$, then*

$$|\mathbb{E}_Q[r] - \mathbb{E}_Q[r_\tau]| \leq e^{-\tau}(1 - Se^{-K}).$$

*Proof.* We establish the upper bounds by finding a worst case $r$ to find the largest $|\mathbb{E}_Q[r] - \mathbb{E}_Q[r_\tau]|$. First, without loss of generality, we may assume that $r(\boldsymbol{x}) \in (-\infty, e^{-\tau}] \cup [e^\tau, \infty)$ for all $\boldsymbol{x} \in \mathcal{X}$. Otherwise, denote $\mathcal{X}_\tau(r) = \{\boldsymbol{x} \in \mathcal{X} : e^{-\tau} < r(\boldsymbol{x}) < e^\tau\}$ as the (measurable) set where the $r(\boldsymbol{x})$ values are between $e^{-\tau}$ and $e^\tau$. Let

$$V_\tau(r) = \int_{\boldsymbol{x} \in \mathcal{X}_\tau(r)} r(\boldsymbol{x}) \mathrm{d}\boldsymbol{x} \in (e^{-\tau}|\mathcal{X}_\tau(r)|, e^\tau|\mathcal{X}_\tau(r)|) \tag{39}$$

be the integral of $r$ over $\mathcal{X}_\tau(r)$. We can transform $r(\boldsymbol{x})$ for all $\boldsymbol{x} \in \mathcal{X}_\tau(r)$ to have values only in $\{e^{-\tau}, e^\tau\}$ and still integrate to $V_\tau(r)$, so the expectation under $Q$ is not changed.

Then we show that we can rescale all the values above $e^\tau$ and below $e^\tau$ to the same value without changing the expected value under $Q$. We denote

$$K_1 = \log \int I(r(\boldsymbol{x}) \leq e^{-\tau})r(\boldsymbol{x})\mathrm{d}Q(\boldsymbol{x}) - \log \int I(r(\boldsymbol{x}) \leq e^{-\tau})\mathrm{d}Q(\boldsymbol{x}) \tag{40}$$

$$K_2 = \log \int I(r(\boldsymbol{x}) \geq e^\tau)r(\boldsymbol{x})\mathrm{d}Q(\boldsymbol{x}) - \log \int I(r(\boldsymbol{x}) \geq e^\tau)\mathrm{d}Q(\boldsymbol{x}) \tag{41}$$

where $e^{K_1}$ and $e^{K_2}$ represents the mean of $r(\boldsymbol{x})$ for all $r(\boldsymbol{x}) \leq e^{-\tau}$ and $r(\boldsymbol{x}) \geq e^\tau$ respectively. We then have:

$$\mathbb{E}_Q[r] = e^{K_1} \int I(r(\boldsymbol{x}) \leq e^{-\tau})\mathrm{d}Q(\boldsymbol{x}) + e^{K_2} \int I(r(\boldsymbol{x}) \geq e^\tau)\mathrm{d}Q(\boldsymbol{x}) \tag{42}$$

$$1 = \int I(r(\boldsymbol{x}) \leq e^{-\tau})\mathrm{d}Q(\boldsymbol{x}) + \int I(r(\boldsymbol{x}) \geq e^\tau)\mathrm{d}Q(\boldsymbol{x}) \tag{43}$$

so we can parametrize $\mathbb{E}_Q[r]$ via $K_1$ and $K_2$. Since $E_Q[r] = S$ by assumption, we have:

$$\int I(r(\boldsymbol{x}) \leq e^{-\tau})\mathrm{d}Q(\boldsymbol{x}) = \frac{e^{K_2} - S}{e^{K_2} - e^{-K_1}} \tag{44}$$

and from the definition of $r_\tau(\boldsymbol{x})$:

$$\mathbb{E}_Q[r_\tau] = \frac{e^{K_2}e^{-\tau} - Se^{-\tau} + Se^\tau - e^{-K_1}e^\tau}{e^{K_2} - e^{-K_1}} := g(K_1, K_2) \tag{45}$$

We can obtain an upper bound once we find $\max g(K_1, K_2)$ and $\min g(K_1, K_2)$. First, we have:

$$\frac{\partial g(K_1, K_2)}{\partial K_1} = \frac{e^{-K_1}e^\tau(e^{K_2} - e^{-K_1}) - e^{-K_1}(e^{K_2}e^{-\tau} - Se^{-\tau} + Se^\tau - e^{-K_1}e^\tau)}{(e^{K_2} - e^{-K_1})^2}$$

$$= \frac{e^{-K_1}(e^\tau - e^{-\tau})(e^{K_2} - S)}{(e^{K_2} - e^{-K_1})^2} \geq 0 \tag{46}$$

$$\frac{\partial g(K_1, K_2)}{\partial K_2} = \frac{e^{K_2}e^{-\tau}(e^{K_2} - e^{-K_1}) - e^{K_2}(e^{K_2}e^{-\tau} - Se^{-\tau} + Se^\tau - e^{-K_1}e^\tau)}{(e^{K_2} - e^{-K_1})^2}$$

$$= \frac{e^{K_2}(e^\tau - e^{-\tau})(e^{-K_1} - S)}{(e^{K_2} - e^{-K_1})^2} \leq 0 \tag{47}$$

Therefore, $g(K_1, K_2)$ is largest when $K_1 \to \infty, K_2 = \tau$ and smallest when $K_1 = \tau, K_2 \to \infty$.

$$\max g(K_1, K_2) = \lim_{K \to \infty} \frac{1 - e^{-K}e^{\tau} + S(e^{\tau} - e^{-\tau})}{e^{\tau} - e^{-K}} = S + e^{-\tau} - Se^{-2\tau} \tag{48}$$

$$\min g(K_1, K_2) = \lim_{K \to \infty} \frac{e^{K}e^{-\tau} - 1 + S(e^{\tau} - e^{-\tau})}{e^{K} - e^{-\tau}} = e^{-\tau} \tag{49}$$

Therefore,

$$|\mathbb{E}_Q[r] - \mathbb{E}_Q[r_\tau]| \le \max(|\max g(K_1, K_2) - S|, |S - \min g(K_1, K_2)|) \tag{50}$$

$$= \max\left(|e^{-\tau} - Se^{-2\tau}|, |S - e^{-\tau}|\right) \tag{51}$$

The proof for Theorem 3 simply follows the above analysis for fixed $K$. When $\tau < K$, we consider the case when $K_1 \to \infty, K_2 = \tau$ and $K_1 = \tau, K_2 = K$; when $\tau > K$ only the smaller values will be clipped, so the increased value is no larger than the case where $K_1 \to \infty, K_2 = K$:

$$\frac{e^{K} - S}{e^{K}} \cdot e^{\tau} = e^{-\tau}(1 - Se^{-K}) \tag{52}$$

where $e^{K} \ge S$ from the fact that $\int r \, dQ = S$. $\qquad \square$

**Theorem 4.** *The variance of the estimator $\mathbb{E}_{Q_n}[r_\tau]$ (using $n$ samples from Q) satisfies:*

$$\text{Var}[\mathbb{E}_{Q_n}[r_\tau]] \le \frac{e^{\tau} - e^{-\tau}}{4n} \tag{18}$$

*Proof.* Since $r_\tau(\boldsymbol{x})$ is bounded between $e^{\tau}$ and $e^{-\tau}$, we have

$$\text{Var}[r_\tau] \le \frac{e^{\tau} - e^{-\tau}}{4} \tag{53}$$

Taking the mean of $n$ independent random variables gives us the result. $\qquad \square$

Combining Theorem 3 and 4 with the bias-variance trade-off argument, we have the following:

**Corollary 3.** *Let $r(\boldsymbol{x}) : \mathcal{X} \to \mathbb{R}_{\ge 0}$ be any non-negative measurable function such that $\int r dQ = S$, $S \in (0, \infty)$ and $r(\boldsymbol{x}) \in [0, e^{K}]$. Define $r_\tau(\boldsymbol{x}) = \text{clip}(r(\boldsymbol{x}), e^{\tau}, e^{-\tau})$ for finite, non-negative $\tau$ and $\mathbb{E}_{Q_n}$ as the sample average of $n$ i.i.d. samples from Q. If $\tau < K$, then*

$$\mathbb{E}_Q[(r - \mathbb{E}_{Q_n}[r_\tau])^2] \le \max\left(e^{-\tau}|1 - Se^{-\tau}|, \left|\frac{1 - e^{K}e^{-\tau} + S(e^{K} - e^{\tau})}{e^{K} - e^{-\tau}}\right|\right)^2 + \frac{e^{\tau} - e^{-\tau}}{4n};$$

*If $\tau \ge K$, then:*

$$\mathbb{E}_Q[(r - \mathbb{E}_{Q_n}[r_\tau])^2] \le e^{-2\tau}(1 - Se^{-K})^2 + \frac{e^{\tau} - e^{-\tau}}{4n} \tag{54}$$

## B  ADDITIONAL EXPERIMENTAL DETAILS

### B.1  BENCHMARK TASKS

**Tasks** We sample each dimension of $(\boldsymbol{x}, \boldsymbol{y})$ independently from a correlated Gaussian with mean $0$ and correlation of $\rho$, where $\mathcal{X} = \mathcal{Y} = \mathbb{R}^{20}$. The true mutual information is computed as:

$$I(\boldsymbol{x}, \boldsymbol{y}) = -\frac{d}{2} \log \left( 1 - \frac{\rho}{2} \right) \tag{55}$$

The initial mutual information is 2, and we increase the mutual information by 2 every $4k$ iterations, so the total training iterations is $20k$.

**Architecture and training procedure** For all the *discriminative* methods, we consider two types of architectures – *joint* and *separable*. The *joint* architecture concatenates the inputs $\boldsymbol{x}, \boldsymbol{y}$, and then passes through a two layer MLP with 256 neurons in each layer with ReLU activations at each layer. The *separaable architecture* learns two separate neural networks for $\boldsymbol{x}$ and $\boldsymbol{y}$ (denoted as $g(\boldsymbol{x})$ and $h(\boldsymbol{y})$) and predicts $g(\boldsymbol{x})^\top h(\boldsymbol{y})$; $g$ and $h$ are two neural networks, each is a two layer MLP with 256 neurons in each layer with ReLU activations at each layer; the output of $g$ and $h$ are 32 dimensions.

For the generative method, we consider the invertible flow architecture described in (Dinh et al., 2014; 2016). $p_\theta, p_\phi, p_\psi$ are flow models with 5 coupling layers (with scaling), where each layer contains a neural network with 2 layers of 100 neurons and ReLU activation. For all the cases, we use with the Adam optimizer (Kingma & Ba, 2014) with learning rate $5 \times 10^{-4}$ and $\beta_1 = 0.9, \beta_2 = 0.999$ and train for $20k$ iterations with a batch size of 64, following the setup in Poole et al. (2019).

**Additional results** We show the bias, variance and mean squared error of the *discriminative* approaches in Table 2. We include additional results for $I_{\text{SMILE}}$ with $\tau = 10.0$.

| | | Gaussian | | | | | Cubic | | | | |
|---|---|---|---|---|---|---|---|---|---|---|---|
| | MI | 2 | 4 | 6 | 8 | 10 | 2 | 4 | 6 | 8 | 10 |
| Bias | CPC | 0.25 | 0.99 | 2.31 | 4.00 | 5.89 | 0.72 | 1.48 | 2.63 | 4.20 | 5.99 |
| | NWJ | 0.12 | 0.30 | 0.75 | 2.30 | 2.97 | 0.66 | 1.21 | 2.04 | 3.21 | 4.70 |
| | SMILE ($\tau = 1.0$) | 0.15 | 0.30 | 0.32 | 0.18 | 0.03 | 0.47 | 0.77 | 1.16 | 1.64 | 2.16 |
| | SMILE ($\tau = 5.0$) | 0.13 | 0.11 | 0.19 | 0.54 | 0.86 | 0.71 | 1.22 | 1.55 | 1.84 | 2.16 |
| | SMILE ($\tau = 10.0$) | 0.14 | 0.21 | 0.22 | 0.11 | 0.19 | 0.70 | 1.28 | 1.83 | 2.44 | 3.02 |
| | SMILE ($\tau = \infty$) | 0.15 | 0.21 | 0.22 | 0.12 | 0.22 | 0.71 | 1.29 | 1.82 | 2.35 | 2.81 |
| | GM (Flow) | 0.11 | 0.14 | 0.15 | 0.14 | 0.17 | 1.02 | 0.47 | 1.85 | 2.93 | 3.55 |
| Var | CPC | 0.04 | 0.04 | 0.02 | 0.01 | 0.00 | 0.03 | 0.04 | 0.03 | 0.01 | 0.01 |
| | NWJ | 0.06 | 0.22 | 1.36 | 16.50 | 99.0 | 0.04 | 0.10 | 0.41 | 0.93 | 3.23 |
| | SMILE ($\tau = 1.0$) | 0.05 | 0.12 | 0.20 | 0.28 | 0.34 | 0.04 | 0.10 | 0.14 | 0.20 | 0.30 |
| | SMILE ($\tau = 5.0$) | 0.05 | 0.11 | 0.19 | 0.31 | 0.51 | 0.04 | 0.07 | 0.12 | 0.18 | 0.26 |
| | SMILE ($\tau = 10.0$) | 0.05 | 0.13 | 0.31 | 0.69 | 1.35 | 0.03 | 0.10 | 0.21 | 0.46 | 0.79 |
| | SMILE ($\tau = \infty$) | 0.05 | 0.14 | 0.36 | 0.75 | 1.54 | 0.03 | 0.12 | 0.24 | 0.65 | 0.94 |
| | GM (Flow) | 0.05 | 0.10 | 0.13 | 0.16 | 0.19 | 0.56 | 0.72 | 0.92 | 1.02 | 1.02 |
| MSE | CPC | 0.10 | 1.02 | 5.33 | 16.00 | 34.66 | 0.55 | 2.22 | 6.95 | 17.62 | 35.91 |
| | NWJ | 0.07 | 0.32 | 2.19 | 33.37 | 28.43 | 0.47 | 1.55 | 4.56 | 11.13 | 27.00 |
| | SMILE ($\tau = 1.0$) | 0.08 | 0.21 | 0.30 | 0.32 | 0.31 | 0.26 | 0.69 | 1.49 | 2.90 | 4.98 |
| | SMILE ($\tau = 5.0$) | 0.07 | 0.13 | 0.22 | 0.57 | 1.26 | 0.54 | 1.56 | 2.53 | 3.58 | 4.92 |
| | SMILE ($\tau = 10.0$) | 0.07 | 0.18 | 0.36 | 0.67 | 1.33 | 0.52 | 1.75 | 3.54 | 6.41 | 9.91 |
| | SMILE ($\tau = \infty$) | 0.08 | 0.19 | 0.40 | 0.76 | 1.62 | 0.54 | 1.75 | 3.55 | 6.09 | 8.81 |
| | GM (Flow) | 0.07 | 0.11 | 0.14 | 0.17 | 0.22 | 1.65 | 0.91 | 4.36 | 9.70 | 13.67 |

Table 2: Bias, Variance and MSE of the estimators under the joint critic.

We show the bias, variance and MSE results in Figure 7. We also evaluate the variance of estimating $\mathbb{E}_{Q_n}[r_\tau]$ (partition function with clipped ratios) for different values of $\tau$ in the SMILE estimator in Figure 8b. With smaller $\tau$ we see a visible decrease in terms of variance in this term; this is consistent with the variance estimates in Figure 7, as there the variance of $\mathbb{E}_{P_n}[\log r]$ is also considered.

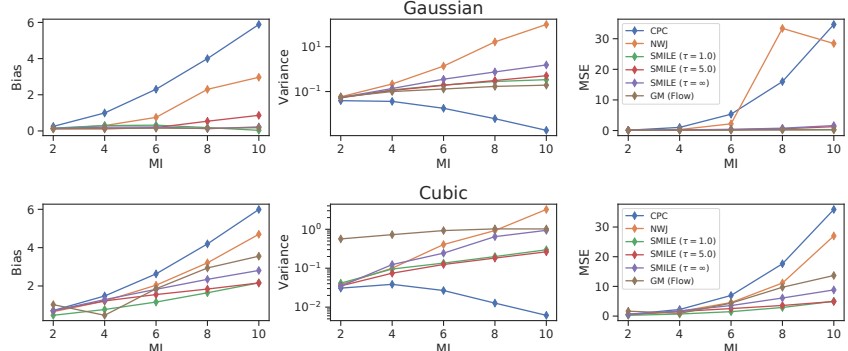

Figure 7: Bias / Variance / MSE of various estimators. on **Gaussian** (top) and **Cubic** (down).

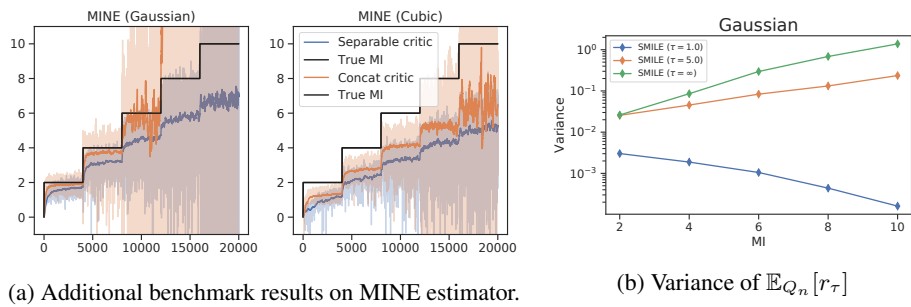

(a) Additional benchmark results on MINE estimator.

(b) Variance of $\mathbb{E}_{Q_n}[r_\tau]$

Figure 8: Additional benchmark results.

### B.2  SELF-CONSISTENCY EXPERIMENTS

**Tasks**   We consider three tasks with the mutual information estimator $\hat{I}$:

1. $\hat{I}(X; Y)$ where $X$ is an image from MNIST (LeCun et al., 1998) or CIFAR10 (Krizhevsky et al., 2012) and $Y$ is the top $t$ rows of $X$. To simplify architecture designs, we simply mask out the bottom rows to be zero, see Figure 3.

2. $\hat{I}([X, X]; [Y; h(Y)])$ where $X$ is an image, $Y$ is the top $t$ rows of $X$, $h(Y)$ is the top $(t - 3)$ rows of $Y$ and $[\cdot, \cdot]$ denotes concatenation. Ideally, the prediction should be close to $\hat{I}(X; Y)$.

3. $\hat{I}([X_1, X_2], [Y_1, Y_2])$ where $X_1$ and $X_2$ are independent images from MNIST or CIFAR10, $Y_1$ and $Y_2$ are the top $t$ rows of $X_1$ and $X_2$ respectively. Ideally, this prediction should be close to $2 \cdot \hat{I}(X; Y)$.

**Architecture and training procedure**   We consider the same architecture for all the *discriminative* approaches. The first layer is a convolutional layer with 64 output channels, kernel size of 5, stride of 2 and padding of 2; the second layer is a convolutional layer with 128 output channels, kernel size of 5, stride of 2 and padding of 2. This is followed another fully connected layer with 1024 neurons and finally a linear layer that produces an output of 1. All the layers (except the last one) use ReLU activations. We stack variables over the channel dimension to perform concatenation.

For the generative approach, we consider the following VAE architectures. The encoder architecture is identical to the *discriminative approach* except the last layer has 20 outputs that predict the mean and standard deviations of 10 Gaussians respectively. The decoder for MNIST is a two layer MLP with 400 neurons each; the decoder for CIFAR10 is the corresponding transposed convolution network for the encoder. All the layers (except the last layers for encoder and decoder) use ReLU activations. For concatenation we stack variables over the channel dimension. For all the cases, we use with the Adam optimizer (Kingma & Ba, 2014) with learning rate $10^{-4}$ and $\beta_1 = 0.9, \beta_2 = 0.999$.

For $I_{\mathrm{GM}}$ we train for 10 epochs, and for the *discriminative* methods, we train for 2 epochs, due to numerical stability issues of $I_{\mathrm{MINE}}$.

**Additional experiments on scaling, rotation and translation**    We consider additional benchmark experiments on MNIST where instead of removing rows, we apply alternative transformations such as random scaling, rotation and translations. For random scaling, we upscale the image randomly by 1x to 1.2x; for random rotation, we randomly rotate the image between $\pm 20$ degrees; for random translation, we shift the image randomly by no more than 3 pixels horizontally and vertically. We consider evaluating the data processing and additivity properties, where the ideal value for the former is no more than 1, and the ideal value for the latter is 2. From the results in Table 3, none of the considered approaches achieve good results in all cases.

|  |  | CPC | MINE | GM (VAE) | SMILE ($\tau = 5.0$) | SMILE ($\tau = \infty$) |
|---|---|---|---|---|---|---|
| | Scaling | 1.00 | 1.03 | 1.12 | 1.19 | 1.04 |
| Data-Processing | Rotation | 1.00 | 1.30 | 1.13 | 1.03 | 1.27 |
| | Translation | 1.00 | 1.28 | 1.01 | 1.07 | 1.08 |
| | Scaling | 1.00 | 1.55 | 1.89 | 1.04 | 1.18 |
| Additivity | Rotation | 1.00 | 2.09 | 1.58 | 1.50 | 1.78 |
| | Translation | 1.00 | 1.41 | 1.28 | 1.32 | 1.33 |

Table 3: Self-consistency experiments on other image transforms.

