# OpenReview forum: "Understanding the Limitations of Variational Mutual Information Estimators"
_ICLR.cc/2020/Conference — Accept (Poster)_

### Official Review · AnonReviewer2 · 2019-10-21
**Official Blind Review #2**

**Rating:** 6

**Review:**

This paper relates most existing variational mutual information estimators to density ratio estimation and uses this to show high variance of certain categories of estimators; this motivates a new SMILE estimator that uses clipping to explicitly control the variance of the estimator (while potentially increasing bias). The paper also conducts some experiments on how well mutual information estimators satisfy some of the core properties of mutual information in practice.

The core result of Theorem 2, although is proof is very simple once the problem has been phrased appropriately, is enlightening.

The self-consistency tests are also sensible and important, although it would be nice to understand better which of these properties are truly important for various uses of mutual information in the literature.

The proposed SMILE estimator is natural. But:

- As $\tau \to \infty$, the variance bound of Theorem 4 naturally becomes infinite (as we'd expect). But we'd hope that the bias would become 0, when in fact Theorem 3 only shows in this case that the bias is at most S. Of course it should be true that $\mathbb E_Q[r] = \mathbb E_Q[r_\infty]$; can we fix Theorem 3 so that it recovers this fact?

Then the biggest question not really addressed by this paper is: how do you set $\tau$?

- If we know the (approximate) value of $S$, then we can presumably trade off between the bias and variance terms of Theorem 3 and 4 to find the most effective value for $\tau$. (Indeed, it might be nice to write a corollary showing that if you set $\tau$ to something depending on $n$ and some upper bound on $S$, you achieve some rate in L2 risk.)

- Is it possible to relate the value of $S$ to, say, the suboptimality in cross-entropy loss of the classifier used to obtain $r$ (when getting $r$ that way)?

- Is it possible to get a probabilistic upper bound on $S$, either with some other technique or perhaps a simple Hoeffding/Bernstein-type bound on the SMILE estimator with some initial value of $\tau$, and then use that to choose the rate for $\tau$?

- Is there some other practical scheme (even if it doesn't admit bounds) to set $\tau$ to control the bias, given that it depends on $S$?



Smaller points and minor suggestions:

- In the paragraph following Theorem 2, it is first of all strange to say that "alternative procedures to obtaining the density ratio estimates will not alleviate the high variance issue" when indeed your proposal does just that. The last sentence of this paragraph clarifies the issue, but still makes $r \ne r^*$ sound "bad," when in fact it's a trade-off; this could be rephrased more clearly.

- In Theorem 3: I think writing $e^{-2 \tau} \lvert S - e^\tau \rvert$ would be clearer than $\lvert e^{-\tau} - S e^{-2 \tau} \rvert$ as you have it.

- In the multivariate Gaussian experiments, Section 6.1: why do you effectively warm-start the estimator for higher mutual informations? Is it just so you can put things in one plot per method? I think it would be more natural to train each different problem from scratch; you can still use the same plot format if you like (with additional vertical lines between problems or something), but it's not clear whether the behavior of the estimator is affected by this choice.

- Why are the flow-based results not shown in Figure 2 / Table 2?

- In Section 6.2 when setting up the three evaluation settings, it would be good to have a forward reference to Figure 3 (since these settings are perhaps easier to understand visually).

**Experience Assessment:**

I have read many papers in this area.

**Review Assessment: Checking Correctness Of Derivations And Theory:**

I assessed the sensibility of the derivations and theory.

**Review Assessment: Checking Correctness Of Experiments:**

I assessed the sensibility of the experiments.

**Review Assessment: Thoroughness In Paper Reading:**

I read the paper at least twice and used my best judgement in assessing the paper.

---

### Official Review · AnonReviewer1 · 2019-10-23
**Official Blind Review #1**

**Rating:** 6

**Review:**

This work summarizes the existing methods of mutual information estimation in a variational inference framework and describes the limitations in terms of bias-variance tradeoffs. Further, the authors care about the self-consistency, namely, independence, data processing, and additivity, which are properties of both entropy and differential entropy. Further, density ratio clipping is proposed to lessen a high-variance problem in estimating a partition function.

In regard to the clipped density rations, it is not clear why both variance and bias become small when S is close to 1 and using small $\tau$ in Theorem 3 and 4. In Fig. 2 on the benchmark experiment, $\tau=5.0$ showed lower variance than $\tau=1.0$.

The comparison with MINE and BA is also expected on the benchmark experiment.

As for the self-consistency tests on images, especially, for the ‘data-processing’ property, it would better to apply affine transformation such as rotation, translation, and scaling, rather than rows deletion.

In Section 3.1, $\Gamma$ is not defined.


**Experience Assessment:**

I have read many papers in this area.

**Review Assessment: Checking Correctness Of Derivations And Theory:**

I did not assess the derivations or theory.

**Review Assessment: Checking Correctness Of Experiments:**

I assessed the sensibility of the experiments.

**Review Assessment: Thoroughness In Paper Reading:**

I read the paper at least twice and used my best judgement in assessing the paper.

---

### Official Review · AnonReviewer3 · 2019-10-25
**Official Blind Review #3**

**Rating:** 6

**Review:**

Summary:
As the title suggests the paper focusses mostly on a negative result:
Mutual information (MI) estimators obtained by variational methods have severe
limitations that make them potentially not useful for down stream tasks.
Besides highlighting the problems with variational MI estimators the authors
suggest a modification to slightly improve the performance of MI estimators
based on partition functions by reducing their variance when MI is high.

The authors give a good overview / introduction of various approaches to
variational MI estimation by discriminative and generative methods.
Generally, MI estimation involves the estimation of the KL divergence between
the joint distribution and the product of the marginals.
The authors present a unifying view on the different approaches that optimizes
the log density ratio required for the KL divergence over the space of
log density ratios.
Discriminative approaches model the density ratio directly (through e.g. neural
network models) and generative approaches model the separate densities
(as generative models where it is possible to evaluate the (conditional)
probabilities / likelihoods of the data generating process).

The authors prove that discriminative approaches that are based on the partition
function approach suffer from high variance where mutual information is high
(Theorem 2).
The estimator based on a finite sample has high variance even if the density
ratio approximation is correct.
(The partition function approach is a way of staying constrained to the log
density ratio function space.)
This high variance problem is something that has previously been observed
empirically and is the main theoretical point that is being made about
limitations of MI estimators.

In order to slightly alleviate the problem of high variance the authors suggest
a way of biasing MI estimators by clipping the density ratio estimates
through a constant chosen as a hyper-parameter.
They prove that their clipping approach reduces variance and therefore
introduces a bias variance tradeoff.

In their later experiments the clipped version of the discriminative approach
performs much better in terms of variance than without clipping and also better
than a generative approach.

In order to empirically evaluate the quality of MI estimators the authors
suggest three criteria that they call self-consistency:
(i) independence, (ii) data processing, (iii) additivity

Self-consistency is evaluated experimentally on images where mutual information
is computed between original image and image with part covered.

The authors claim and experimentally show that discriminative approaches fail
in (iii) and generative approaches fail in (i), (ii).
Overall, variational MI approaches do not satisfy self-consistency.

Evaluation:
I suggest to accept the paper.
The theoretical contribution of showing the variance limitation of
discriminative approaches seems significant.
That insight leads to the idea that clipping can be a useful bias that
significantly reduces variance without making the already biased anyways
results much worst in the experiments.
However, I also feel like the paper is not yet as focused as it could be.
It contains many concepts that could need a little bit more space.

Suggestions:
- Page 2: Nitpick, but in the definition of $L^p(Q)$ using $\colon$ twice is
	not super readable on the first read

- Page 2: In the definition of $I_{BA}$ clearify whether $p(x)$ is a marginal
	or a joint density (as $P$ is the cumulative joint)

- Page 3: In Theorem 1 what is the definition of $P \ll Q$?
	This suggests an order on the space of measures $\mathcal P$?

- Page 3: "Obtain an density ratio estimate" -> Obtain a density ratio estimate


**Experience Assessment:**

I do not know much about this area.

**Review Assessment: Checking Correctness Of Derivations And Theory:**

I did not assess the derivations or theory.

**Review Assessment: Checking Correctness Of Experiments:**

I did not assess the experiments.

**Review Assessment: Thoroughness In Paper Reading:**

I made a quick assessment of this paper.

---

### Decision · Program_Chairs · 2019-12-19

**Decision:**

Accept (Poster)

**Comment:**

This paper presents a critical appraisal of variational mutual information estimators, and suggests a slight variance-reducing improvement based on clipping density ratio estimates, and prove that this reduces variance (at the cost of bias). They also propose a set of criteria they term "self-consistency" for evaluation of MI estimators and, and show convincingly that variational MI estimators fall short with respect to these.

Reviewers were generally positive about the contribution, and were happy with improvements made. While somewhat limited in scope, I believe this is nonetheless a valuable contribution to the conversation surrounding mutual information objectives that have become popular recently. I therefore recommend acceptance.